# FourierFormer: Transformer Meets Generalized Fourier Integral Theorem

**Tan M. Nguyen**[*]
Department of Mathematics
University of California, Los Angeles
`tanmnguyen89@ucla.edu`

**Minh Pham**[*]
Department of Mathematics
University of California, Los Angeles
`minhrose@ucla.edu`

**Tam Nguyen**
Department of ECE
Rice University
`nguyenminhtam9520@gmail.com`

**Khai Nguyen**
Department of Statistics and Data Sciences
University of Texas at Austin
`khainb@utexas.edu`

**Stanley J. Osher**[**]
Department of Mathematics
University of California, Los Angeles
`sjo@math.ucla.edu`

**Nhat Ho**[**]
Department of Statistics and Data Sciences
University of Texas at Austin
`minhnhat@utexas.edu`

## Abstract

Multi-head attention empowers the recent success of transformers, the state-of-the-art models that have achieved remarkable success in sequence modeling and beyond. These attention mechanisms compute the pairwise dot products between the queries and keys, which results from the use of unnormalized Gaussian kernels with the assumption that the queries follow a mixture of Gaussian distribution. There is no guarantee that this assumption is valid in practice. In response, we first interpret attention in transformers as a nonparametric kernel regression. We then propose the FourierFormer, a new class of transformers in which the dot-product kernels are replaced by the novel generalized Fourier integral kernels. Different from the dot-product kernels, where we need to choose a good covariance matrix to capture the dependency of the features of data, the generalized Fourier integral kernels can automatically capture such dependency and remove the need to tune the covariance matrix. We theoretically prove that our proposed Fourier integral kernels can efficiently approximate any key and query distributions. Compared to the conventional transformers with dot-product attention, FourierFormers attain better accuracy and reduce the redundancy between attention heads. We empirically corroborate the advantages of FourierFormers over the baseline transformers in a variety of practical applications including language modeling and image classification.

## 1 Introduction

Transformers [83] are powerful neural networks that have achieved tremendous success in many areas of machine learning [40, 76, 36] and become the state-of-the-art model on a wide range of applications across different data modalities, from language [23, 1, 18, 13, 62, 4, 8, 21] to images [24, 43, 78, 63, 59, 27], videos [3, 44], point clouds [97, 31], and protein sequence [65, 34]. In addition to their excellent performance on supervised learning tasks, transformers can also effectively transfer the learned knowledge from a pretraining task to new tasks with limited or no supervision [60, 61, 23, 94, 42]. At the core of transformers is the dot-product self-attention, which

---

[*] Co-first authors. [**] Co-last authors. Please correspond to: tanmnguyen89@ucla.edu

36th Conference on Neural Information Processing Systems (NeurIPS 2022).

mainly accounts for the success of transformer models [14, 56, 41]. This dot-product self-attention learn self-alignment between tokens in an input sequence by estimating the relative importance of a given token with respect to all other tokens. It then transform each token into a weighted average of the feature representations of other tokens where the weight is proportional to a importance score between each pair of tokens. The importance scores in self-attention enable a token to attend to other tokens in the sequence, thus capturing the contextual representation [6, 83, 38].

## 1.1 Self-Attention

Given an input sequence $\boldsymbol{X} := [\boldsymbol{x}_1, \cdots, \boldsymbol{x}_N]^\top \in \mathbb{R}^{N \times D_x}$ of $N$ feature vectors, self-attention computes the output sequence $\mathbf{H}$ from $\boldsymbol{X}$ as follows:

**Step 1: Projecting the input sequence into different subspaces.** The input sequence $\boldsymbol{X}$ is transformed into the query matrix $\mathbf{Q}$, the key matrix $\mathbf{K}$, and the value matrix $\mathbf{V}$ via three linear transformations

$$\mathbf{Q} = \boldsymbol{X}\mathbf{W}_Q^\top; \mathbf{K} = \boldsymbol{X}\mathbf{W}_K^\top; \mathbf{V} = \boldsymbol{X}\mathbf{W}_V^\top,$$

where $\mathbf{W}_Q, \mathbf{W}_K \in \mathbb{R}^{D \times D_x}$, and $\mathbf{W}_V \in \mathbb{R}^{D_v \times D_x}$ are the weight matrices. We denote $\boldsymbol{Q} := [\boldsymbol{q}_1, \cdots, \boldsymbol{q}_N]^\top, \mathbf{K} := [\boldsymbol{k}_1, \cdots, \boldsymbol{k}_N]^\top$, and $\mathbf{V} := [\mathbf{v}_1, \cdots, \mathbf{v}_N]^\top$, where the vectors $\boldsymbol{q}_i, \boldsymbol{k}_i, \mathbf{v}_i$ for $i = 1, \cdots, N$ are the query, key, and value vectors, respectively.

**Step 2: Computing the output as a weighted average.** The output sequence $\mathbf{H} := [\boldsymbol{h}_1, \cdots, \boldsymbol{h}_N]^\top$ is then given by

$$\mathbf{H} = \mathrm{softmax}\left(\mathbf{Q}\mathbf{K}^\top/\sqrt{D}\right)\mathbf{V} := \mathbf{A}\mathbf{V}, \tag{1}$$

where the softmax function is applied to each row of the matrix $(\mathbf{Q}\mathbf{K}^\top)/\sqrt{D}$. For each query vector $\boldsymbol{q}_i, i = 1, \cdots, N$, Eqn. (1) can be written in the vector form to compute the output vector $\boldsymbol{h}_i$ as follows

$$\boldsymbol{h}_i = \sum_{j=1}^{N} \mathrm{softmax}\left(\boldsymbol{q}_i^\top \boldsymbol{k}_j/\sqrt{D}\right)\mathbf{v}_j := \sum_{j=1}^{N} a_{ij}\mathbf{v}_j. \tag{2}$$

The matrix $\mathbf{A} \in \mathbb{R}^{N \times N}$ and its component $a_{ij}$ for $i, j = 1, \cdots, N$ are the attention matrix and attention scores, respectively. The self-attention computed by equations (1) and (2) is called the dot-product attention or softmax attention. In our paper, we refer a transformer that uses this attention as the baseline transformer with the dot-product attention or the dot-product transformer. The structure of the attention matrix $\mathbf{A}$ after training governs the ability of the self-attention to capture contextual representation for each token.

**Multi-head Attention** Each output sequence $\mathbf{H}$ forms an attention head. Multi-head attention concatenates multiple heads to compute the final output. Let $H$ be the number of heads and $\mathbf{W}^O \in \mathbb{R}^{HD_v \times HD_v}$ be the projection matrix for the output. The multi-head attention is defined as

$$\mathrm{MultiHead}(\{\mathbf{Q}, \mathbf{K}, \mathbf{V}\}_{i=1}^{H}) = \mathrm{Concat}(\mathbf{H}_1, \ldots, \mathbf{H}_H)\mathbf{W}^O.$$

The capacity of the attention mechanism and its ability to learn diverse syntactic and semantic relationships determine the success of transformers [77, 84, 17, 85, 32]. However, equations (1) and (2) implies that the dot-product attention assumes the features $(q_{i1}, \ldots, q_{iD})$ in $\boldsymbol{q}_i$, as well as the features $(k_{j1}, \ldots, q_{jD})$ in $\boldsymbol{k}_j$, are independent. Thus, the dot-product attention fail to capture the correlations between these features, limiting its representation capacity and inhibit the performance of transformers on practical tasks where there is no guarantee that independent features can learned from complex data. One solution to capture correlations between features $\boldsymbol{q}_i$ and $\boldsymbol{k}_j$ is to introduce covariance matrices into the formulation of the dot-product attention with the cost of significantly increasing of the computational complexity. Also, choosing good covariance matrices is difficult.

## 1.2 Contribution

In this paper, we first establish a correspondence between self-attention and nonparametric kernel regression. Under this new perspective of self-attention, we explain the limitation of the dot-product self-attention that it may fail to capture correlations between the features in the query and key vectors. We then leverage the generalized Fourier integral theorems, which can automatically capture these correlations, and derive the generalized Fourier integral estimators for the nonparametric regression problem. Using this new density estimator, we propose the FourierFormer, a novel class of transformers that can capture correlations between features in the query and key vectors of self-attention. In summary, our contribution is three-fold:

1. We derive the formula of self-attention from solving a nonparametric kernel regression problem, thus providing a nonparametric regression interpretation to study and further develop self-attention.

2. We develop the generalized Fourier integral estimators for the nonparametric regression problem and provide theoretical guarantees for these estimator.

3. We propose the FourierFormer whose attentions use the generalized Fourier integral estimators to capture more efficiently correlations between features in the query and key vectors.

Finally, we empirically show that the FourierFormer attains significantly better accuracy than the baseline transformer with the dot-product attention on a variety of tasks including the WikiText language modeling and ImageNet image classsification. We also demonstrate in our experiments that FourierFormer helps reduce the redundancy between attention heads.

**Organization** We structure this paper as follows: In Section 2, we present the correspondence between self-attention and nonparametric kernel regression. In Section 3, we discuss the generalized Fourier integral estimators and define the FourierFormer. We validate and empirically analyze the advantages of FourierFormer in Section 4. We discuss related works in Section 5. The paper ends with concluding remarks. Technical proofs and more experimental details are provided in the Appendix.

**Notation** For any $N \in \mathbb{N}$, we denote $[N] = \{1, 2, \ldots, N\}$. For any $D \geq 1$, $\mathbb{L}_1(\mathbb{R}^D)$ denotes the space of real-valued functions on $\mathbb{R}^D$ that are integrable. For any two sequences $\{a_N\}_{N \geq 1}, \{b_N\}_{N \geq 1}$, we denote $a_N = \mathcal{O}(b_N)$ to mean that $a_N \leq C b_N$ for all $N \geq 1$ where $C$ is some universal constant.

## 2   A Nonparametric Regression Interpretation of Self-attention

In this section, we establish the connection between self-attention and nonparametric kernel regression. In particular, we derive the self-attention in equation (2) as a nonparametric kernel regression in which the key vectors $\boldsymbol{k}_j$ and value vectors $\mathbf{v}_j$ are training inputs and training targets, respectively, while the query vectors $\boldsymbol{q}_i$ and the output vectors $\boldsymbol{h}_i$ form a set of new inputs and their corresponding targets that need to be estimated, respectively, for $i, j = 1, \cdots, N$. In general, we can view the training set $\{\boldsymbol{k}_j, \mathbf{v}_j\}$ for $j \in [N]$ to come from the following *nonparametric regression model*:

$$\mathbf{v}_j = f(\boldsymbol{k}_j) + \varepsilon_j, \tag{3}$$

where $\varepsilon_1, \ldots, \varepsilon_N$ are independent noises such that $\mathbb{E}(\varepsilon_j) = 0$. Furthermore, we consider a random design setting where the key vectors $\boldsymbol{k}_1, \boldsymbol{k}_2, \ldots, \boldsymbol{k}_N$ are i.i.d. samples from the distribution that admits $p$ as density function. By an abuse of notation, we also denote $p$ as the joint density where the key and value vectors $(\mathbf{v}_1, \boldsymbol{k}_1), \ldots, (\mathbf{v}_N, \boldsymbol{k}_N)$ are i.i.d. samples from. Here, $f$ is a true but unknown function and we would like to estimate it.

**Nadaraya–Watson estimator** Our approach to estimate the function $f$ is based on Nadaraya–Watson's nonparametric kernel regression approach [50]. In particular, from the nonparametric regression model (3), we have $\mathbb{E}[\mathbf{v}_j | \boldsymbol{k}_j] = f(\boldsymbol{k}_j)$ for all $j \in [N]$. Therefore, it is sufficient to estimate the conditional distribution of the value vectors given the key vectors. Given the density function $p$ of the key vectors and the joint density $p$ of the key and value vectors, for any pair of vectors $(\mathbf{v}, \boldsymbol{k})$ generate from model (3) we have

$$\mathbb{E}[\mathbf{v}|\boldsymbol{k}] = \int_{\mathbb{R}^D} \mathbf{v} \cdot p(\mathbf{v}|\boldsymbol{k}) d\mathbf{v} = \int \frac{\mathbf{v} \cdot p(\mathbf{v}, \boldsymbol{k})}{p(\boldsymbol{k})} d\mathbf{v}. \tag{4}$$

The formulation (4) of the conditional expectation indicates that as long as we can estimate the joint density function $p(\mathbf{v}, \boldsymbol{k})$ and the marginal density function $p(\mathbf{v})$, we are able to obtain an estimation for the conditional expectation and thus for the function $f$. This approach is widely known as Nadaraya–Watson's nonparametric kernel regression approach.

**Kernel density estimator** To estimate $p(\mathbf{v}, \boldsymbol{k})$ and $p(\boldsymbol{k})$, we employ the kernel density estimation approach [66, 57]. In particular, by using the isotropic Gaussian kernel with bandwidth $\sigma$, we have the following estimators of $p(\mathbf{v}, \boldsymbol{k})$ and $p(\boldsymbol{k})$:

$$\hat{p}_\sigma(\mathbf{v}, \boldsymbol{k}) = \frac{1}{N} \sum_{j=1}^N \varphi_\sigma(\mathbf{v} - \mathbf{v}_j) \varphi_\sigma(\boldsymbol{k} - \boldsymbol{k}_j), \qquad \hat{p}_\sigma(\boldsymbol{k}) = \frac{1}{N} \sum_{j=1}^N \varphi_\sigma(\boldsymbol{k} - \boldsymbol{k}_j), \tag{5}$$

where $\varphi_\sigma(.)$ is the isotropic multivariate Gaussian density function with diagonal covariance matrix $\sigma^2 \mathbf{I}_D$. Given the kernel density estimators (5), we obtain the following estimation of the function $f$:

$$
\begin{aligned}
\widehat{f}_\sigma(\boldsymbol{k}) &= \int_{\mathbb{R}^D} \frac{\mathbf{v} \cdot \hat{p}_\sigma(\mathbf{v}, \boldsymbol{k})}{\hat{p}_\sigma(\boldsymbol{k})} d\mathbf{v} = \int_{\mathbb{R}^D} \frac{\mathbf{v} \cdot \sum_{j=1}^N \varphi_\sigma(\mathbf{v} - \mathbf{v}_j) \varphi_\sigma(\boldsymbol{k} - \boldsymbol{k}_j)}{\sum_{j=1}^N \varphi_\sigma(\boldsymbol{k} - \boldsymbol{k}_j)} d\mathbf{v} \\
&= \frac{\sum_{j=1}^N \varphi_\sigma(\boldsymbol{k} - \boldsymbol{k}_j) \int \mathbf{v} \cdot \varphi_\sigma(\mathbf{v} - \mathbf{v}_j) d\mathbf{v}}{\sum_{j=1}^N \varphi_\sigma(\boldsymbol{k} - \boldsymbol{k}_j)} = \frac{\sum_{j=1}^N v_j \varphi_\sigma(\boldsymbol{k} - \boldsymbol{k}_j)}{\sum_{j=1}^N \varphi_\sigma(\boldsymbol{k} - \boldsymbol{k}_j)}.
\end{aligned}
\tag{6}
$$

**Connection between Self-Attention and nonparametric regression** By plugging the query vectors $\boldsymbol{q}_i$ into the function $\widehat{f}_\sigma$ in equation (6), we obtain that

$$
\begin{aligned}
\widehat{f}_\sigma(\boldsymbol{q}_i) &= \frac{\sum_j^N \mathbf{v}_j \exp\left(-\|\boldsymbol{q}_i - \boldsymbol{k}_j\|^2 / 2\sigma^2\right)}{\sum_j^N \exp\left(-\|\boldsymbol{q}_i - \boldsymbol{k}_j\|^2 / 2\sigma^2\right)} \\
&= \frac{\sum_j^N \mathbf{v}_j \exp\left[-\left(\|\boldsymbol{q}_i\|^2 + \|\boldsymbol{k}_j\|^2\right)/2\sigma^2\right] \exp\left(\boldsymbol{q}_i \boldsymbol{k}_j^\top / \sigma^2\right)}{\sum_j^N \exp\left[-\left(\|\boldsymbol{q}_i\|^2 + \|\boldsymbol{k}_{j'}\|^2\right)/2\sigma^2\right] \exp\left(\boldsymbol{q}_i \boldsymbol{k}_j^\top / \sigma^2\right)}.
\end{aligned}
\tag{7}
$$

If we further assume that the keys $\boldsymbol{k}_j$ are normalized, which is usually done in practice to stabilize the training of transformers [71], the value of $\widehat{f}_\sigma(\boldsymbol{q}_i)$ in equation (6) then becomes

$$
\widehat{f}_\sigma(\boldsymbol{q}_i) = \frac{\sum_j^N \mathbf{v}_j \exp\left(\boldsymbol{q}_i \boldsymbol{k}_j^\top / \sigma^2\right)}{\sum_j^N \exp\left(\boldsymbol{q}_i \boldsymbol{k}_j^\top / \sigma^2\right)} = \sum_{j=1}^N \mathrm{softmax}\left(\boldsymbol{q}_i^\top \boldsymbol{k}_j / \sigma^2\right) \mathbf{v}_j.
\tag{8}
$$

When we choose $\sigma^2 = \sqrt{D}$ where $D$ is the dimension of $\boldsymbol{q}_i$ and $\boldsymbol{k}_j$, equation (8) matches equation (2) of self-attention, namely, $\widehat{f}_\sigma(\boldsymbol{q}_i) = \boldsymbol{h}_i$. Thus, we have shown that self-attention performs nonparametric regression using isotropic Gaussian kernels.

**Remark 1** *The assumption that $\boldsymbol{k}_j$ is normalized is to recover the pairwise dot-product attention in transformers. In general, this assumption is not necessary. In fact, the isotropic Gaussian kernel in equation (7) is more desirable than the dot-product kernel in equation (8) of the pairwise dot-product attention since the former is Lipschitz while the later is not Lipschitz [37]. The Lipschitz constraint helps improve the robustness of the model [16, 81, 2] and stabilize the model training [48].*

**Limitation of Self-Attention** From our nonparametric regression interpretation, self-attention is derived from the use of isotropic Gaussian kernels for kernel density estimation and nonparametric regression estimation, which may fail to capture the complex correlations between $D$ features in $\boldsymbol{q}_i$ and $\boldsymbol{k}_j$ [88, 33]. Using multivariate Gaussian kernels with dense covariance matrices can help capture such correlations; however, choosing good covariance matrices is challenging and inefficient [87, 73, 11]. In the following section, we discuss the Fourier integral estimator and its use as a kernel for computing self-attention in order to overcome these limitations.

## 3 FourierFormer: Transformer via Generalized Fourier Integral Theorem

In the following, we introduce generalized integral theorems that are able to capture the complex interactions among the features of the queries and keys. We then apply these theorems to density estimation and nonparametric regression problems. We also establish the convergence rates of these estimators. Given these density estimators, we introduce a novel family of transformers, named *FourierFormer*, that integrates the generalized Fourier integral theorem into the dot-product attention step of the standard transformer.

### 3.1 Generalized Fourier Integral Theorems and Their Applications

The Fourier integral theorem is a beautiful result in mathematics [92, 7] and has been recently used in nonparametric mode clustering, deconvolution problem, and generative modeling [33]. It is a combination of Fourier transform and Fourier inverse transform. In particular, for any function

$p \in \mathbb{L}_1(\mathbb{R}^D)$, the *Fourier integral theorem* is given by

$$p(\boldsymbol{k}) = \frac{1}{(2\pi)^D} \int_{\mathbb{R}^D} \int_{\mathbb{R}^D} \cos(\boldsymbol{s}^\top (\boldsymbol{k} - \boldsymbol{y})) p(\boldsymbol{y}) d\boldsymbol{y} d\boldsymbol{s}$$

$$= \frac{1}{\pi^D} \lim_{R \to \infty} \int_{\mathbb{R}^D} \prod_{j=1}^D \frac{\sin(R(k_j - y_j))}{(k_j - y_j)} p(\boldsymbol{y}) d\boldsymbol{y}, \tag{9}$$

where $\boldsymbol{k} = (k_1, \ldots, k_D), \boldsymbol{y} = (y_1, \ldots, y_D), \boldsymbol{s} = (s_1, \ldots, s_D)$, and $R$ is the radius. The detailed derivation of Equation (9) is in Appendix B.3. Equation (9) suggests that $p_R(\boldsymbol{k}) := \frac{1}{\pi^D} \int_{\mathbb{R}^D} \prod_{j=1}^D \frac{\sin(R(y_j - k_j))}{(y_j - k_j)} p(\boldsymbol{y}) d\boldsymbol{y}$ can be used as an estimator of the function $p$.

**Benefits of the Fourier integral over Gaussian kernel** There are two important benefits of the estimator $p_R$: (i) it can automatically preserve the correlated structure lying within $p$ even when $p$ is very complex and high dimensional function. It is in stark contrast to the standard kernel estimator built based on multivariate Gaussian kernel where we need to choose good covariance matrix in the multivariate Gaussian kernel to guarantee such estimator to work well. We note that as the standard soft-max Transformer is constructed based on the multivariate Gaussian kernel, the issue of choosing good covariance matrix in dot-product transformer is inevitable; (ii) The product of sinc kernels in the estimator $p_R$ does not decay to a point mass when $R \to \infty$. It is in stark difference from the multivariate Gaussian kernel estimator, which converges to a point mass when the covariance matrix goes to 0. It indicates that $p_R$ is a non-trivial estimator of the function $p$. Finally, detailed illustrations of these benefits of the Fourier integral over Gaussian kernel in density estimation and nonparametric regression problems, which we have just shown to have connection to the self-attention in transformer, can be found in Section 8 in [33].

**Generalized Fourier integral estimator** Borrowing the above benefits of Fourier integral estimator $p_R$, in the paper we would like to consider a generalization of that estimator, named *generalized Fourier integral estimator*, which is given by:

$$p_R^\phi(\boldsymbol{k}) := \frac{R^D}{A^D} \int_{\mathbb{R}^D} \prod_{j=1}^D \phi\left(\frac{\sin(R(y_j - k_j))}{R(y_j - k_j)}\right) p(\boldsymbol{y}) d\boldsymbol{y}, \tag{10}$$

where $A := \int_{\mathbb{R}} \phi\left(\frac{\sin(z)}{z}\right) dz$ and $\phi : \mathbb{R} \to \mathbb{R}$ is a given function. When $\phi(\boldsymbol{k}) = \boldsymbol{k}$ for all $\boldsymbol{k} \in \mathbb{R}^D$, the generalized Fourier integral estimator $p_R^\phi$ becomes the Fourier integral estimator $p_R$. Under appropriate conditions on the function $\phi$ (see Theorem 1 in Section 3.1.1 and Theorem 3 in Appendix C.1), the estimator $p_R^\phi$ converges to the true function $p$, namely,

$$p(\boldsymbol{k}) = \lim_{R \to \infty} p_R^\phi(\boldsymbol{k}) = \lim_{R \to \infty} \frac{R^D}{A^D} \int_{\mathbb{R}^D} \prod_{j=1}^D \phi\left(\frac{\sin(R(y_j - k_j))}{R(y_j - k_j)}\right) p(\boldsymbol{y}) d\boldsymbol{y}. \tag{11}$$

We name the above limit as *generalized Fourier integral theorem*. Furthermore, the estimator $p_R^\phi$ also inherits similar aforementioned benefits of the Fourier integral estimator $p_R$. Therefore, we will use the generalized Fourier integral theorem as a building block for constructing density estimators and nonparametric regression estimators, which are crucial to develop the FourierFormer in Section 3.2.

### 3.1.1 Density Estimation via Generalized Fourier Integral Theorems

We first apply the generalized Fourier integral theorem to the density estimation problem. To ease the presentation, we assume that $\boldsymbol{k}_1, \boldsymbol{k}_2, \ldots, \boldsymbol{k}_N \in \mathbb{R}^D$ are i.i.d. samples from a distribution admitting density function $p$ where $D \geq 1$ is the dimension. Inspired by the generalized Fourier integral theorem, we obtain the following *generalized Fourier density estimator* $p_{N,R}^\phi$ of $p$ as follows:

$$p_{N,R}^\phi(\boldsymbol{k}) := \frac{R^D}{NA^D} \sum_{i=1}^N \prod_{j=1}^D \phi\left(\frac{\sin(R(k_j - k_{ij}))}{R(k_j - k_{ij})}\right), \tag{12}$$

where $A = \int_{\mathbb{R}} \phi\left(\frac{\sin(z)}{z}\right) dz$ and $\boldsymbol{k}_i = (k_{i1}, \ldots, k_{iD})$ for all $i \in [N]$. To quantify the error between the generalized Fourier density estimator $p_{n,R}^\phi$ and the true density $p$, we utilize mean integrated

squared errors (MISE) [91], which is given by:

$$\text{MISE}(p_{N,R}^{\phi}, p) := \int_{\mathbb{R}^D} (p_{N,R}^{\phi}(\boldsymbol{k}) - p(\boldsymbol{k}))^2 d\boldsymbol{k}. \tag{13}$$

We start with the following bound on the MISE between $p_{n,R}^{\phi}$ and $p$.

**Theorem 1** *Assume that $\int_{\mathbb{R}} \phi(\sin(z)/z)z^j dz = 0$ for all $j \in [m]$ and $\int_{\mathbb{R}} |\phi(\sin(z)/z)||z|^{m+1} dz < \infty$ for some $m \in \mathbb{N}$. Then, there exist universal constants $C$ and $C'$ depending on $d$ and $A$ such that*

$$\text{MISE}(p_{N,R}^{\phi}, p) \leq \frac{C}{R^{m+1}} + \frac{C'R^D}{N}.$$

Proof of Theorem 1 is in Appendix D.1. A few comments are in order. First, by choosing $R$ to balance the bias and variance in the bound of MISE in Theorem 1, we have the optimal $R$ as $R = \mathcal{O}(N^{1/(D+m+1)})$. With that choice of $R$, the MISE rate of $p_{N,R}^{\phi}$ is $\mathcal{O}(N^{-(m+1)/(D+m+1)})$. Second, when $\phi(z) = z^l$ for $l \geq 4$ and $z \in \mathbb{R}$, the assumptions in Theorem 1 are satisfied when $m = 1$. Under this case, the MISE rate of $p_{N,R}^{\phi}$ is $\mathcal{O}(N^{-2/(D+2)})$. However, these assumptions do not satisfy when $\phi(z) = z^l$ and $l \in \{1, 2, 3\}$, which is due to the limitation of the current proof technique of Theorem 1 that is based on Taylor expansion of the estimator $p_{n,R}^{\phi}$.

To address the limitation of the Taylor expansion technique, we utilize the Plancherel theorem in Fourier analysis to establish the MISE rate of $p_{N,R}^{\phi}$ when $\phi(z) = z^l$ and $l \in \{1, 2, 3\}$. The details of the theoretical analyses for such setting are in Appendix C.

## 3.2 FourierFormer: Transformers with Fourier Attentions

Motivated by the preservation of the correlated structure of the function from the generalized Fourier integral theorem as well as the theoretical guarantees of density estimators, in this section we adapt the nonparametric regression interpretation of self-attention in Section 2 and propose the generalized Fourier nonparametric regression estimator in Section 3.2.1. We also establish the convergence properties of that estimator. Then, based on generalized Fourier nonparametric regression estimator, we develop the Fourier Attention and its corresponding FourierFormer in Section 3.2.2.

### 3.2.1 Nonparametric Regression via Generalized Fourier Integral Theorem

We now discuss an application of the generalized Fourier integral theorems to the nonparametric regression setting (3), namely, we assume that $(\mathbf{v}_1, \boldsymbol{k}_1), \ldots, (\mathbf{v}_N, \boldsymbol{k}_N)$ are i.i.d. samples from the following nonparametric regression model:

$$\mathbf{v}_j = f(\boldsymbol{k}_j) + \varepsilon_j,$$

where $\varepsilon_1, \ldots, \varepsilon_N$ are independent noises such that $\mathbb{E}(\varepsilon_j) = 0$ and the key vectors $\boldsymbol{k}_1, \boldsymbol{k}_2, \ldots, \boldsymbol{k}_N$ are i.i.d. samples from $p$. Given the generalized Fourier density estimator (12), following the argument in Section 2, the Nadaraya–Watson estimator of the function $f$ based on the generalized Fourier density estimator is given by:

$$f_{N,R}(\boldsymbol{k}) := \frac{\sum_{i=1}^{N} \mathbf{v}_i \prod_{j=1}^{D} \phi\left(\frac{\sin(R(k_j - k_{ij}))}{R(k_j - k_{ij})}\right)}{\sum_{i=1}^{N} \prod_{j=1}^{D} \phi\left(\frac{\sin(R(k_j - k_{ij}))}{R(k_j - k_{ij})}\right)}. \tag{14}$$

The main difference between the generalized Fourier nonparametric regression estimator $f_{N,R}$ in equation (14) and the estimator $\widehat{f}_{\sigma}$ in equation (6) is that the estimator $f_{N,R}$ utilizes the generalized Fourier density estimator to estimate the conditional distribution of the value vectors given the key vectors instead of the isotropic Gaussian kernel density estimator as in $\widehat{f}_{\sigma}$. As we highlighted in Section 3, an important benefit of the generalized Fourier density estimator is that it can capture the complex dependencies of the features of the value vectors and the key vectors while the Gaussian kernel needs to have good covariance matrix to do that, which is computationally expensive in practice.

We now have the following result establishing the mean square error (MSE) of $f_{N,R}$ when $D_v = 1$.

**Theorem 2** *Assume that $\int_{\mathbb{R}} \phi\left(\frac{\sin(z)}{z}\right) z^j dz = 0$ for all $1 \leq j \leq m$ and $\int_{\mathbb{R}} \left|\phi\left(\frac{\sin(z)}{z}\right)\right| |z|^j dz < \infty$ for any $m+1 \leq j \leq 2m+2$ for some $m \in \mathbb{N}$. Then, for any $\boldsymbol{k} \in \mathbb{R}^D$, when $D_v = 1$ there exist universal constants $C_1, C_2, C_3, C_4$ such that the following holds:*

$$\mathbb{E}\left[(f_{N,R}(\boldsymbol{k}) - f(\boldsymbol{k}))^2\right] \leq \left(\frac{C_1}{R^{2(m+1)}} + \frac{(f(\boldsymbol{k}) + C_2)R^D}{N}\right) \Big/ \left(p^2(\boldsymbol{k})J(R)\right),$$

*where $J(R) = 1 - \frac{1}{p^2(\boldsymbol{k})}\left(\frac{C_3}{R^{2(m+1)}} + \frac{C_4 R^d \log(NR)}{N}\right)$. Here, the outer expectation is taken with respect to the key vectors $\boldsymbol{k}_1, \ldots, \boldsymbol{k}_N$ and the noises $\varepsilon_1, \ldots, \varepsilon_N$.*

Proof of Theorem 2 is in Appendix D.3. A few comments with Theorem 2 are in order. First, by choosing $R$ to balance the bias and variance in the bound of the MSE of the nonparametric generalized Fourier estimator $f_{N,R}$, we have the optimal radius $R$ as $R = \mathcal{O}(N^{\frac{1}{2(m+1)+D}})$. With that choice of the optimal radius $R$, the rate of $f_{N,R}$ is $\mathcal{O}(N^{-\frac{2(m+1)}{D+2(m+1)}})$. Second, when $\phi(z) = z^l$ for $l \geq 6$, the assumption on the function $\phi$ of Theorem 2 is satisfied with $m = 1$. Under this case, the rate of $f_{N,R}$ becomes $\mathcal{O}(N^{-\frac{4}{D+4}})$. In Appendix C, we also provide the rate of $f_{N,R}$ when $\phi(z) = z^l$ for some $l \leq 5$, which includes the original Fourier integral theorem.

### 3.2.2 FourierFormer

Given the generalized Fourier nonparametric regression estimator $f_{N,R}$ in equation (14), by plugging the query values $\boldsymbol{q}_1, \ldots, \boldsymbol{q}_N$ into that function, we obtain the following definition of the Fourier attention:

**Definition 1 (Fourier Attention)** *A Fourier attention is a multi-head attention that does nonparametric regression using the generalized Fourier nonparametric regression estimator $f_{N,R}$. The output $\hat{\boldsymbol{h}}_i$ of the Fourier attention is then computed as*

$$\hat{\boldsymbol{h}}_i := f_{N,R}(\boldsymbol{q}_i) = \frac{\sum_{i=1}^{N} \mathbf{v}_i \prod_{j=1}^{D} \phi\left(\frac{\sin(R(q_{ij}-k_{ij}))}{R(q_{ij}-k_{ij})}\right)}{\sum_{i=1}^{N} \prod_{j=1}^{D} \phi\left(\frac{\sin(R(q_{ij}-k_{ij}))}{R(q_{ij}-k_{ij})}\right)} \qquad \forall\, i \in [N]. \tag{15}$$

Given the Fourier Attention in Definition 1, we then give the definition of FourierFormer as follows.

**Definition 2 (FourierFormer)** *A FourierFormer is a transformer that uses Fourier attention to capture dependency between tokens in the input sequence and the correlation between features in each token.*

**Remark 2 (The Nonnegativity of the Fourier Kernel)** *The density estimation via generalized Fourier integral theorem in Section 3.1.1 does not require the generalized Fourier density estimator to be nonnegative. However, empirically, we observe that negative density estimator can cause instability in training the FourierFormer. Thus, in FourierFormer, we choose the function $\phi$ to be a nonnegative function to enforce the density estimator to be nonnegative. In particular, we choose $\phi$ to be power functions of the form $\phi(x) = x^{2m}$, where $m$ is an positive integer. Note that when $m = 1$ and $m = 2$, the kernels in our generalized Fourier integral estimators are the well-known Fejer-de la Vallee Poussin and Jackson-de la Vallee Poussin kernels [20].*

### 3.3 An Efficient Implementation of the Fourier Attention

The Fourier kernel is implemented efficiently in the C++/CUDA extension developed by Pytorch [58]. The idea is similar to the function `cdist` [58], which computes the p-norm distance between each pair of the two collections of row vectors. In our case, we aim to compute kernel functions that represent a Fourier attention in Definition 1. The core of this implementation is the following Fourier metric function $d_f$:

$$d_f(\boldsymbol{q}_i, \boldsymbol{k}_j) = \prod_{d=1}^{D} \phi\left(\frac{\sin(R(\boldsymbol{q}_{id} - \boldsymbol{k}_{jd}))}{R(\boldsymbol{q}_{id} - \boldsymbol{k}_{jd})}\right).$$

We directly implement $d_f$ as a `torch.autograd.Function` [58] in which we provide an efficient way to compute forward and backward function ($d_f$ and gradient of $d_f$). While the implementation

**Table 1.** Perplexity (PPL) on WikiText-103 of FourierFormers compared to the baselines. FourierFormers achieve much better PPL than the baselines.

| Method | Valid PPL | Test PPL |
|---|---|---|
| *Baseline dot-product (small)* | 33.15 | 34.29 |
| FourierFormer (small) | **31.86** | **32.85** |
| *Baseline dot-product (medium)* | 27.90 | 29.60 |
| FourierFormer (medium) | **26.51** | **28.01** |

of the forward function is straight forward, the backward function is more tricky since we need to optimize the code to compute the gradient of $d_f$ w.r.t to variables $q$, $k$, and $R$ all at once. We can develop the backward function with highly parallel computation by exploiting GPU architecture and utilizing the reduction technique. The computational time is comparable to function `cdist`; thus, our FourierFormer implementation is as computationally time-efficient.

## 4 Experimental Results

In this section, we numerically justify the advantage of FourierFormer over the baseline dot-product transformer on two large-scale tasks: language modeling on WikiText-103 [46] (Section 4.1) and image classification on ImageNet [22, 67] (Section 4.2), time series classification on the UEA benchmark [5] (Section 4.3), and reinforcement learning on the D4RL Benchmark [29] (Section 4.4), and the machine translation on the IWSLT' 14 De-En [10] (Section 4.5). We aim to show that: (i) FourierFormer achieves better accuracy than the baseline transformer on a variety of practical tasks with different data modalities, and (ii) FourierFormer helps reduce head redundancy compared to the baseline transformer (Section 4.6).

Throughout the section, we compare FourierFormers with the baseline dot-product transformers of the same configuration. In all experiments, we made the constant $R$ in Fourier attention (see equation (16)) to be a learnable scalar and set choose the function $\phi(x) = x^4$ (see Remark 2). All of our results are averaged over 5 runs with different seeds. The details on the models and training are provided in Appendix A. Moreover, additional experiments results are provided in Appendix E. Our PyTorch code with documentation can be found at https://github.com/minhtannguyen/FourierFormer_NeurIPS.

### 4.1 Language Modeling on WikiText-103

We report the validation and test perplexity (PPL) of FourierFormer versus the baseline transformer with the dot-product attention in Table 1. FourierFormers attain much better PPL than the baselines in both small and medium configurations. For the small configuration, the improvements of FourierFormer over the baseline are 1.29 PPL in validation and 1.44 PPL in test. For the medium configuration, these improvements are 1.39 PPL in validation and 1.59 PPL in test. These results suggest that the advantage of FourierFormer over the baseline dot-product transformer grows with the model's size. This meets our expectation because larger models has larger query and key dimensions, e.g. the language model with medium configuration in this experiment has the query and key dimension of 256 versus 128 as in the language model with small configuration. Since the advantage of FourierFormer results from the property that FourierFormer can capture correlation between features in query and key vectors, the larger the query and key dimensions are, the more advantage FourierFormer has.

### 4.2 Image Classification on ImageNet

In the Imagenet classification task, we illustrates the benefits of Fourierformers in different data modalities. We summarize our models' results in Table 2. Same as in the language modeling experiment, for this image classification task, the Deit model equipped with FourierFormer significantly outperforms the baseline Deit dot-product transformer [79] in both top-1 and top-5 accuracy. This result suggests that the advantage of FourierFormer over the baseline dot-product transformer holds across different data modalities.

### 4.3 UEA Time Series Classification

To evaluate Fourierformers on temporal sequences, we compare the accuracy of the our models and the baseline softmax transformers trained on 10 datasets in the the UEA Time Series Classification Archive benchmark [5]. We summarize our results in Table 3. We observe show that Fourierformers outperforms softmax baselines in 7 out of 10 tasks and yields significantly better accuracy than the softmax transformer on average, showing the our models benefits when trained on temporal data.

**Table 2.** Top-1 and top-5 accuracy (%) of FourierFormer Deit vs. the baseline Deit with dot-product attention. FourierFormer Deit outperforms the baseline in both top-1 and top-5 accuracy.

| Method | Top-1 Acc | Top-5 Acc |
|---|---|---|
| *Baseline DeiT* | 72.23 | 91.13 |
| FourierFormer DeiT | **73.25** | **91.66** |

**Table 3.** The FourierFormer vs. the baseline softmax transformer on the UEA Time Series Classification Archive benchmark [5]. The FourierFormer outperforms the baseline. We also include the reported results from [95] and [93] (in parentheses) in addition to our reproduced results. The experiment setups and configurations for the baseline and our FourierFormer are the same as in [93] (for the PEMS-SF, SelfRegulationSCP2, UWaveGestureLibrary datasets) and [95] (for other tasks).

| Dataset/Model | *Baseline softmax* | FourierFormer |
|---|---|---|
| ETHANOLCONCENTRATION | 32.08 (33.70) | **36.12** |
| FACEDETECTION | 68.70 (68.10) | **68.71** |
| HANDWRITING | **32.08 (30.50)** | 31.68 |
| HEARTBEAT | 75.77 (77.60) | **76.42** |
| JAPANESEVOWELS | **99.46 (99.40)** | 99.37 |
| PEMS-SF | 82.66 (82.10) | **86.70** |
| SELFREGULATIONSCP1 | 91.46 (92.50) | **91.70** |
| SELFREGULATIONSCP2 | 54.72 (53.90) | **55.37** |
| SPOKENARABICDIGITS | **99.33 (99.30)** | 99.00 |
| UWAVEGESTURELIBRARY | 84.45 (85.60) | **86.66** |
| AVERAGE ACCURACY | 72.07 (72.27) | **73.17** |

**Table 4.** The decision FourierFormer vs. the baseline decision transformer [12] on the continuous control tasks from D4RL benchmark [29]. The decision FourierFormer yields significantly better results than the baseline decision transformer on 8 out of 9 tasks and on average across tasks. Each experiment result is averaged over 5 runs with different random seeds. We also include the reported results from [93] (in parentheses) in addition to our reproduced results.

| Environment/Model | *Baseline decision transformer* | Decision FourierFormer |
|---|---|---|
| MEDIUM-EXPERT | | |
| HALFCHEETAH | 91.03 (83.80) | **92.27** |
| HOPPER | 110.30 (104.40) | **111.10** |
| WALKER | 108.70 (107.70) | **108.90** |
| MEDIUM-REPLAY | | |
| HALFCHEETAH | 35.31 (34.6) | **38.47** |
| HOPPER | 85.61 (79.70) | **89.70** |
| WALKER | **66.11 (62.90)** | 63.19 |
| MEDIUM | | |
| HALFCHEETAH | 42.28 (42.40) | **42.38** |
| HOPPER | 61.47 (64.20) | **64.77** |
| WALKER | 68.68 (70.60) | **70.42** |
| AVG REWARD | 74.39 (72.20) | **75.69** |

### 4.4 Reinforcement learning on the D4RL benchmark

We also examine the performance of our Fourierformers in reinforcement learning. In Table 4, we verify the advantage of decision FourierFormer over the baseline decision transformer [12] on the continuous control tasks from the D4RL benchmark [29]. The decision FourierFormer is the decision transformer with the Fourier attention instead of the softmax attention. On this benchmark, our decision FourierFormer significantly outperforms the baseline decision transformer on 8 out of 9 tasks and on average across tasks. Each experiment result averaged over 5 runs with different random seeds. We follow the architecture and training configuration from [93].

### 4.5 Machine Translation on IWSLT' 14 De-En

We demonstrate the performance of Fourierformer on the IWSLT' 14 De-En [10] neural machine translation task, which has different inputs' the sequence lengths. Table 5 shows that the FourierFormer achieves better BLUE scores than the softmax baseline.

### 4.6 FourierFormer Helps Reducing Head Redundancy

To study the diversity between attention heads, given the model trained for the WikiText-103 language modeling task, we compute the average $\mathcal{L}_2$ distance between heads in each layer. We show the layer-average mean and variance of distances between heads in Table 6. Results in Table 6 shows

**Table 5.** The FourierFormer vs. the baseline softmax transformer on the IWSLT'14 De-En machine translation benchmark [10]. The FourierFormer outperforms the baseline.

| Method | BLEU score |
|---|---|
| *Baseline softmax* | 34.42 |
| FourierFormer | **34.68** |

**Table 6.** Layer-average mean and standard deviation of $\mathcal{L}_2$ distances between heads of FourierFormer versus the baseline transformer with dot-product attention trained for the WikiText-103 language modeling task. FourierFormer has greater $\mathcal{L}_2$ distance between heads than the baseline and thus captures more diverse attention patterns.

| Method | Mean | Variance |
|---|---|---|
| *Baseline dot-product* | $6.20 \pm 2.30$ | $6.17 \pm 2.30$ |
| FourierFormer | $\mathbf{7.45 \pm 2.50}$ | $\mathbf{7.37 \pm 2.44}$ |

that FourierFormer obtains greater $\mathcal{L}_2$ distance between attention heads than the baseline transformer with the dot-product attention and thus helps reduce the head redundancy. Note that we use the small configuration as specified in Section 4.1 for both models.

## 5    Related Work

**Interpretation of Attention Mechanism in Transformers** Recent works have tried to gain an understanding of transformer's attention from different perspectives. [80] considers attention as applying kernel smoother over the inputs. Extending this kernel approach, [35, 15, 52, 89, 54] linearize the softmax kernel in dot-product attention and propose a family of efficient transformers with linear computational and memory complexity. [9] then shows that these linear transformers are comparable to a Petrov-Galerkin projection [64], suggesting that the softmax normalization in the dot-product attention is sufficient but not necessary. Other works provide an understanding of attention in transformers via ordinary/partial differential equation include [45, 69]. In addition, [51, 75, 30, 96, 53] relate attentions in transformers to a Gaussian mixture models. Several works also connect the attention mechanism to graph-structured learning and message passing in graphical models [90, 72, 39]. Our work focuses on deriving the connection between self-attention and nonparametric kernel regression and exploring better regression estimator, such as the generalized Fourier nonparametric regression estimator, to improve the performance of transformers.

**Redundancy in Transformers** [19, 47, 25] show that neurons and attention heads in the pre-trained transformer are redundant and can be removed when applied on a downstream task. By studying the contextualized embeddings in pre-trained networks, it has been demonstrated that the learned representations from these redundant models are highly anisotropic [49, 26]. Furthermore, [70, 74, 86, 68] employ knowledge distillation and sparse approximation to enhance the efficiency of transformers. Our FourierFormer is complementary to these methods and can be combined with them.

## 6    Concluding Remarks

In this paper, we establish the correspondence between the nonparametric kernel regression and the self-attention in transformer. We then develop the generalized Fourier integral estimators and propose the FourierFormer, a novel class of transformers that use the generalized Fourier integral estimators to construct their attentions for efficiently capturing the correlations between features in the query and key vectors. We theoretically prove the approximation guarantees of the generalized Fourier integral estimators and empirically validate the advantage of FourierFormer over the baseline transformer with the dot-product attention in terms of accuracy and head redundancy reduction. It is interesting to incorporate robust kernels into the nonparametric regression framework of FourierFormer to enhance the robustness of the model under data perturbation and adversarial attacks. A limitation of FourierFormer is that it still has the same quadratic computational and memory complexity as the baseline transformer with the dot-product attention. We leave the development of the linear version of FourierFormer that achieves linear computational and memory complexity as future work. It is worth noting that there is no potential negative societal impacts of FourierFormer.

## Acknowledgements

This material is based on research sponsored by the AFOSR MURI FA9550-18-1-0502, the ONR grant N00014-20-1-2093, the MURI N00014-20-1-2787, and the NSF under Grant# 2030859 to the Computing Research Association for the CIFellows Project (CIF2020-UCLA-38). NH acknowledges support from the NSF IFML 2019844 and the NSF AI Institute for Foundations of Machine Learning.

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
