# OpenReview forum: "FourierFormer: Transformer Meets Generalized Fourier Integral Theorem"
_NeurIPS.cc/2022/Conference — NeurIPS 2022 Accept_

### Official Review · Reviewer_WPHc · 2022-07-11

**Rating:** 6
**Confidence:** 4
**Soundness:** 4 excellent
**Presentation:** 3 good
**Contribution:** 3 good

**Summary:**

This paper proposes the FourierFormer, in which the dot-product kernels are replaced by the generalized Fourier integral kernels. Unlike the dot-product kernels, where we need to choose a good covariance matrix to capture the dependency of the features of data, the generalized Fourier integral kernels can automatically capture such dependency and remove the need to tune the covariance matrix. This paper theoretically prove that the proposed Fourier integral kernels can efficiently approximate key and query distributions and verify this point through experiments on two transformer-based tasks.

**Questions:**

1. In line 144, the subsection title is “(Generalized) Fourier Integral Theorems”. Why do authors bracket the word “generalized” ？


**Limitations:**

The paper didn't address the limitation and potential negative societal impact of the work.

**Strengths And Weaknesses:**

1.This paper introduces a new angle to interpret transformer and its key module. This work provides a nonparametric regression interpretation to study self-attention in transformers and formulate self-attention from the viewpoint of kernel regression.
2.This work adopts the generalized Fourier integral estimators to replace the traditional dot-product self-attention and provide theoretical guarantees for the estimator.
3.Overall, the paper is well organized and technically sound. The experimental results on multiple transformer-based tasks verify the efficiency of the proposed Fourier Former.

Weaknesses
1.The derivation process and the presentation need to be improved. Some important symbol annotations or explanation is missing during the algorithm description, which make readers hard to follow the derivation process. For example, in the equation (9), some important symbol annotations are missing, e.g. ‘s’, ‘R’. It is difficult for readers to catch up the derivation, and the derivation of p(k) is crucial to the following interpretation.
2.Some pitfalls in the paper: a) in line 100, “are i.i.d samples from.” ; b) in equation (6) one of /psi is written as /phi; c)line 185, the C’ in text are in wrong format.

---

> ### Author Response · Authors · 2022-08-02
> **Response to Reviewer WPHc (1)**
>
> Thank you for your thoughtful review and valuable feedback. Below we address your concerns.
>
> -----
>
> **Q1. The derivation process and the presentation need to be improved. Some important symbol annotations or explanation is missing during the algorithm description, which make readers hard to follow the derivation process. For example, in the equation (9), some important symbol annotations are missing, e.g. ‘s’, ‘R’. It is difficult for readers to catch up the derivation, and the derivation of p(k) is crucial to the following interpretation.**
>
> **Reply:** Thanks for your suggestions and pointing out the missing annotations. We have revised the derivation in our revision. We have also  added a background section on the nonparametric kernel regression, kernel density estimation, and the Fourier Integral theorem in Appendix A of the revision so that the readers can easily follow the derivation.
>
> **Q2. Some pitfalls in the paper: a) in line 100, “are i.i.d samples from.” ; b) in equation (6) one of /psi is written as /phi; c)line 185, the C’ in text are in wrong format. In line 144, the subsection title is “(Generalized) Fourier Integral Theorems”. Why do authors bracket the word "generalized"?**
>
> **Reply:** Thanks for pointing these out. We have addressed them in our revision. We have also removed the bracket around the word "Generalized" in the revision.
>
> -----
>
> We hope we have cleared your concerns about our work. We have also revised our manuscript according to your comments, and we would appreciate it if we can get your further feedback at your earliest convenience.

---

> > ### Author Response · Authors · 2022-08-06
> > **Response to Reviewer WPHc - Any further questions on our current draft**
> >
> > We would like to thank you again for your thoughtful reviews and valuable feedback.
> >
> > We would appreciate it if you could let us know if our responses have addressed your concerns and whether you still have any other questions on the current draft and our rebuttal.
> >
> > We would be happy to do any follow-up discussion or address any additional comments.

---

### Official Review · Reviewer_MHeF · 2022-07-11

**Rating:** 6
**Confidence:** 3
**Soundness:** 3 good
**Presentation:** 2 fair
**Contribution:** 3 good

**Summary:**

In this paper, the authors provide a new perspective to interpret the self-attention mechanism in Transformers. In particular, with the assumption that the query and key vectors are normalized, the self-attention mechanism coincides with the well-known Nonparametric Kernel Regression with kernel density estimation. Motivated by this, the authors instead use the Generalized Fourier Integral Theorem to build more powerful estimators for capturing the interaction between features in different dimensions. Experiments on some benchmarks are conducted.

**Questions:**

See the issues in the Strength And Weaknesses. If the authors could address these issues, I would like to increase my scores accordingly.

**Limitations:**

No negative societal impact

**Strengths And Weaknesses:**

**Strengths**
- The interpretation of seeing the self-attention mechanism as using the isotropic Gaussian kernels for kernel density estimation and nonparametric regression estimation seems to be novel, which provides a new perspective to the community to understand the behavior of self-attention.
- The motivation seems to be reasonable to use the generalized Fourier Integral Theorem to capture the feature interaction instead of using the multivariate Gaussian kernels with proper covariance matrices.
- The theoretical analysis is thorough, including approximation error of the generalized Fourier density estimator (Theorem 1) and the generalized Fourier nonparametric regression estimator (Theorem 2).

**Weaknesses**
- **Regarding the background**: the authors should consider adding a preliminary section to introduce the background knowledge on the nonparametric kernel regression, kernel density estimation, and the generalized Fourier Integral theorem, which could help the readers easily follow the derivation of Section 2 and understand the motivation to use the Fourier Integral theorem as a guide to developing a new self-attention mechanism.
- **Regarding the experimental evaluation**: the issues are three-fold. 1) since the authors provide an analysis of the approximation error between estimators and true functions (Theorem 1 and 2), it is informative to provide an empirical evaluation of these quantities on real data as further verification. 2)  The experiments should be more comprehensive and general. For both the language modeling task and image classification task, the model size is limited and the baselines are restrictive. 3) Since the FourierFormer need customized operators for implementation, the authors should also provide the memory/time cost profiling compared to popular Transformer architectures. Based on these issues, the efficiency and effectiveness of the FourierFormer are doubtful.

-------After Rebuttal-------
Thank authors for the detailed response. Most of my concerns have been addressed. I have updated my scores to 6.

---

> ### Author Response · Authors · 2022-08-02
> **Response to Reviewer MHeF (1)**
>
> Thank you for your thoughtful review and valuable feedback. Below we address your concerns.
>
> -----
> **Q1. the authors should consider adding a preliminary section to introduce the background knowledge on the nonparametric kernel regression, kernel density estimation, and the generalized Fourier Integral theorem, which could help the readers easily follow the derivation of Section 2 and understand the motivation to use the Fourier Integral theorem as a guide to developing a new self-attention mechanism.**
>
> **Reply:** Thanks for your suggestion. We have added a background section on the nonparametric kernel regression, kernel density estimation, and the Fourier Integral theorem in Appendix A of the revision. The Generalized Fourier Integral Theorem is our new theorem and a contribution of this paper. We have explained this theorem in detail in Section 3.1 of our paper.
>
> **Q2. Since the authors provide an analysis of the approximation error between estimators and true functions (Theorem 1 and 2), it is informative to provide an empirical evaluation of these quantities on real data as further verification.**
>
> **Reply:** Following the reviewer’s suggestions, we have conducted additional synthetic experiments to empirically justify Theorems 1 and 2. We have described our experiments and summarized our results in Section E.6, Figure 1 and 2 in Appendix E of the revision.

---

> > ### Author Response · Authors · 2022-08-02
> > **Response to Reviewer MHeF (2)**
> >
> > **Q3. The experiments should be more comprehensive and general. For both the language modeling task and image classification task, the model size is limited and the baselines are restrictive.**
> >
> > **Reply:**  First, we respectfully disagree with the reviewer’s comment for the WikiText-103 language modeling task and ImageNet image classification task, the model size is limited and the baselines are restrictive. We use strong baselines for both the WikiText-103 language modeling and the ImageNet image classification task. Particularly, for the WikiText-103 benchmark, our baseline model  with the medium configuration [Schlag et al. (2021)] reported in Table 1 in the paper has 90M parameters, 16 layers, 8 attention heads per layer, and hidden size of 256. The size of our baseline model is close to BERTBase [Devlin et al. (2019)], which has 110M parameters, 12 layers, 12 attention heads per layer, and hidden size of 768. Note that the baseline transformer we used is deeper than BERTBase. This baseline transformer attains a test perplexity (PPL) of 29.60 as reported in [Schlag et al. (2021)], which, on this WikiText-103 task, is better than or equivalent to popular transformer models including  [Grave et al. (2017)], [Dauphin et al. (2017)], [Merity et al. (2018)] and [Rae et al. (2018)]. For the ImageNet benchmark, we use the DeiT-tiny model in [Touvron et al. (2021)]. Even though the name of the model is DeiT-tiny, it is not a small model. The model has 5M parameters, 12 transformer layers, 4 attention heads per layer, and the model dimension of 192. Both the WikiText-103 and ImageNet benchmarks require training the models on 4 NVIDIA 3090Ti GPUs for 3-4 days.
> >
> > Second, as the reviewer suggests, **we have conducted additional experiments** on different benchmarks to justify the advantage of our FourierFormer over the baseline softmax transformer. These additional benchmarks include **IWSLT’ 14 De-En machine translation task** [Cettolo et al. (2014)] and a set of 10 multivariate datasets from the **UEA Time Series Classification Archive** [Bagnall et al. (2018)]. We observe that our FourierFormers outperform the baseline softmax transformers significantly on both of these benchmarks. We summarize our results below and in Table 7 and 8 in Appendix E.4 and E.5, respectively, of the revision.
> >
> > Table 1: The FourierFormer vs. the baseline softmax transformer on the UEA Time Series Classification Archive benchmark
> > | Dataset/Model       | *Baseline softmax (small)*          | Fourierformer   |
> > | :---        |    :----:   |    :----:   |
> > | EthanolConcentration     |   32.08 (33.70)  |   **36.12**    |
> > | FaceDetection     |    68.70 (68.10)   |    **68.71**   |
> > | HandWriting    |    **32.08 (30.50)**    |   31.68    |
> > | HeartBeat    |    75.77 (77.60)    |    **76.42**    |
> > | JapaneseVowels    |    **99.46 (99.40)**  |   99.37   |
> > | PEMS-SF    |    82.66 (82.10)    |   **86.70**   |
> > | SelfRegulationSCP1    |    91.46 (92.50)  |   **91.70**   |
> > | SelfRegulationSCP2   |    54.72 (53.90)   |   **55.37**    |
> > | SpokenArabicDigits   |    **99.33 (99.30)**    |   99.00  |
> > | UWaveGestureLibrary   |    84.45 (85.60)    |  **86.66**  |
> > | Average Accuracy   |   72.07 (72.27)    |  **73.17**   |
> >
> > Table 2: The FourierFormer vs. the baseline softmax transformer on the IWSLT’14 De-En machine translation benchmark
> > | Method      | BLEU score
> > | :---        |    :----:   |
> > | *Baseline softmax*    |   34.42  |
> > | FourierFormer     |    **34.68**   |
> >
> > Also, note that even though we refer to the baselines as the softmax transformers, those baseline models for different benchmarks are different. They use the same softmax attention. In our experiments, we replace this softmax attention in these baselines by the Fourier attention to form the corresponding FourierFormers.
> >
> > **References**
> >
> > [1] Imanol Schlag et al. Linear Transformers are Secretly Fast Weight Programmers. ICML, 2021.
> >
> > ​​[2] Jacob Devlin et al. BERT: Pre-training of Deep Bidirectional Transformers for Language Modeling. NAACL, 2019.
> >
> > [3] Edouard Grave et al. Improving neural language models with a continuous cache. ICLR, 2017.
> >
> > [4] Yann N. Dauphin et al. Language modeling with gated convolutional networks. ICML, 2017.
> >
> > [5] Stephen Merity et al. An analysis of neural language modeling at multiple scales. arXiv, abs/1803.08240, 2018.
> >
> > [6] Jack W. Rae et al. Fast parametric learning with activation memorization. ICML, 2018.
> >
> > [7] Hugo Touvron et al. Training data-efficient image transformers & distillation through attention. ICML, 2021.
> >
> > [8] Mauro Cettolo et al. Report on the 11th iwslt evaluation campaign, iwslt 2014. In Proceedings of the International Workshop on Spoken Language Translation, Hanoi, Vietnam, volume 57, 2014.
> >
> > [9] Anthony Bagnall et al. The UEA multivariate time series classification archive, 2018. arXiv preprint arXiv:1811.00075.

---

> > > ### Author Response · Authors · 2022-08-02
> > > **Response to Reviewer MHeF (3)**
> > >
> > > **Q4. Since the FourierFormer need customized operators for implementation, the authors should also provide the memory/time cost profiling compared to popular Transformer architectures. Based on these issues, the efficiency and effectiveness of the FourierFormer are doubtful.**
> > >
> > > **Reply:** We have included quantitative results on the runtime and GPU memory usage of the FourierFormer versus the baseline softmax transformer in Table 6 in Appendix E.3  of our revision. We summarize our results below. We observe that the runtime of the FourierFormer is slightly more than the runtime of the softmax transformer but can be further reduced with a more optimized implementation. The GPU memory usage of the FourierFormer and the baseline softmax transformer are similar.
> > >
> > > Table 3: Runtime and GPU memory usage of the FourierFormer vs. the baseline softmax transformer. Both models are trained for the WikiText-103 language modeling task
> > > | Model       | Runtime (ms/sample)   (Train)    | GPU Memory (GB) (Train)    |  Runtime (ms/sample)  (Test)    | GPU Memory (GB) (Test)    |
> > > | :---        |    :----:   |    :----:   |   :----:   |    :----:   |
> > > | *Baseline softmax*       |    5.41   |     1.43    |     1.53   |     0.94    |
> > > | FourierFormer   |    6.00    |      1.43    |     1.70   |     0.94    |
> > > -----
> > >
> > >
> > > We hope we have cleared your concerns about our work. We have also revised our manuscript according to your comments, and we would appreciate it if we can get your further feedback at your earliest convenience.

---

> > > > ### Author Response · Authors · 2022-08-06
> > > > **Response to Reviewer MHeF - New Empirical Results to Further Address Q3**
> > > >
> > > > Regarding the concern from the reviewer that the experiments in our paper should be more comprehensive and general, in addition to experiments on IWSLT’ 14 De-En machine translation task [Cettolo et al. (2014)] and the UEA Time Series Classification Archive [Bagnall et al. (2018)] in our previous reply to Q3, as well as on the ImageNet image classification task and the WikiText-103 language modeling task in our paper, we have run additional experiments on the **continuous control tasks from the D4RL benchmark** [Fu et al. (2020)] to evaluate the performance of the FourierFormer on the offline reinforcement learning. On this benchmark, our decision FourierFormer significantly outperforms the baseline decision transformer [Chen et al.(2021)] on 8 out of 9 tasks and on average across tasks. Here, the decision FourierFormer is the decision transformer with the Fourier attention instead of the softmax attention. This result shows that our Fourier attention can be used in different transformer architectures to improve their performance. We include our results below and in Table 9 in Appendix E.7 of our revision. Each experiment result is averaged over 5 runs with different random seeds.
> > > >
> > > > We would appreciate it if you could let us know if our responses have addressed your concerns and whether you still have any other questions on the current draft and our rebuttal. We would be happy to do any follow-up discussion or address any additional comments.
> > > >
> > > >
> > > > Table 4: The decision FourierFormer vs. the baseline decision transformer on the continuous control tasks from D4RL benchmark
> > > >
> > > > | Environment/Model       | Baseline decision transformer          | Decision FourierFormer   |
> > > > | :---        |    :----:   |    :----:   |
> > > > |         |    Medium-Expert   |     |
> > > > | HalfCheetah     |   91.03 (83.80)  |   **92.27**   |
> > > > | Hopper     |    110.30 (104.40)   |    **111.10**  |
> > > > | Walker   |    108.70 (107.70)    |    **108.9**   |
> > > > |         |    Medium-Replay   |     |
> > > > | HalfCheetah     |   35.31 (34.60)  |  **38.47**    |
> > > > | Hopper     |   85.61 (79.70)  |  **89.70**    |
> > > > | Walker     |   **66.11 (62.90)**  |  63.19    |
> > > > |         |    Medium   |     |
> > > > | HalfCheetah     |   42.28 (42.40)  |   **42.38**   |
> > > > | Hopper     |    61.47 (64.20)   |   **64.77**   |
> > > > | Walker   |    68.68 (70.60)    |     **70.42**    |
> > > > | **Avg Reward**   |   74.39 (72.20)    |  **75.69**  |
> > > >
> > > > **References**
> > > >
> > > > [1] Mauro Cettolo et al. Report on the 11th iwslt evaluation campaign, iwslt 2014. In Proceedings of the International Workshop on Spoken Language Translation, Hanoi, Vietnam, volume 57, 2014.
> > > >
> > > > [2] Anthony Bagnall et al. The UEA multivariate time series classification archive. arXiv preprint arXiv:1811.00075, 2018.
> > > >
> > > > [3] Justin Fu et al. D4rl: Datasets for deep data-driven reinforcement learning. arXiv preprint arXiv:2004.07219, 2020.
> > > >
> > > > [4] Lili Chen et al. Decision Transformer: Reinforcement Learning via Sequence Modeling. NeurIPS, 2021.

---

> > > > > ### Comment · Reviewer_MHeF · 2022-08-08
> > > > > **Thanks for the response**
> > > > >
> > > > >  Sincerely thank authors for the detailed responses. Most of my concerns have been addressed and I appreciate the authors' efforts to add background knowledge, additional experimental results, and runtime/memory evaluation. I have updated my rating to weak accept.

---

> > > > > > ### Author Response · Authors · 2022-08-08
> > > > > > **Thanks for your endorsement!**
> > > > > >
> > > > > > Thanks for your response and we appreciate your endorsement.

---

### Official Review · Reviewer_Ky7B · 2022-07-11

**Rating:** 5
**Confidence:** 4
**Soundness:** 2 fair
**Presentation:** 3 good
**Contribution:** 3 good

**Summary:**

The authors demonstrate the FourierFormer, a new class of transformers in which the novel generalized Fourier integral kernels replace the dot-product kernels. The FourierFormer can capture correlations between query features and key self-attention vectors. The authors empirically corroborate the advantages of FourierFormers over the baseline transformers in various practical applications, including language modeling and image classification.

**Questions:**

In the paper, the authors say ‘However, equations (1) and (2) imply that the dot-product attention assumes the features (qi1, . . . , qiD) in qi, as well as the features (kj1, . . . , qjD) in kj , are independent’. This description also implies that v is not included in the consideration of the correlation. However, even it's straightforward to consider q, k as independent variables and v as dependent variables, in my opinion, q, k, and v don’t have a clear statistical relationship in Transformer, partially due to its poor interpretability. Hence, according to the author's idea, v may also be part of considering the correlation.

**Limitations:**

Although the authors have given detailed proof mathematically, due to the poor interpretability of the Transformer itself, I still need to see more experimental results to agree with their point of view. The current experimental results are insufficient and not persuasive.

**Strengths And Weaknesses:**

Strengths:
The ideas that the authors put forward are novel, and the mathematical arguments are complete and ingenious.
Weaknesses:
The experiments in this paper are insufficient and, therefore, not convincing enough to demonstrate the effectiveness of the FourierFormer. The experiment only involves two basic tasks based on WikiText-103 and ImageNet.

---

> ### Author Response · Authors · 2022-08-02
> **Response to Reviewer Ky7B (1)**
>
> Thank you for your thoughtful review and valuable feedback. Below we address your concerns.
>
> -----
>
> **Q1. The experiments in this paper are insufficient and, therefore, not convincing enough to demonstrate the effectiveness of the FourierFormer. The experiment only involves two basic tasks based on WikiText-103 and ImageNet. Although the authors have given detailed proof mathematically, due to the poor interpretability of the Transformer itself, I still need to see more experimental results to agree with their point of view. The current experimental results are insufficient and not persuasive.**
>
> **Reply:** First, we respectfully disagree with the reviewer’s comment that WikiText-103 and ImageNet are basic tasks. Both WikiText-103 and ImageNet are considered large-scale datasets. In particular, WikiText-103 consists of 28K training articles with 103M running words, and ImageNet consists of 1.28M training images. We also use strong baselines for both the WikiText-103 language modeling and the ImageNet image classification task. Particularly, for the WikiText-103 benchmark, our baseline model  with the medium configuration [Schlag et al. (2021)] reported in Table 1 in the paper has 90M parameters, 16 layers, 8 attention heads per layer, and hidden size of 256. The size of our baseline model is close to BERTBase [Devlin et al. (2019)], which has 110M parameters, 12 layers, 12 attention heads per layer, and hidden size of 768. Note that the baseline transformer we used is deeper than BERTBase. This baseline transformer attains a test perplexity (PPL) of 29.60 as reported in [Schlag et al. (2021)], which, on this WikiText-103 task, is better than or equivalent to popular transformer models including  [Grave et al. (2017)], [Dauphin et al. (2017)], [Merity et al. (2018)] and [Rae et al. (2018)]. For the ImageNet benchmark, we use the DeiT-tiny model in [Touvron et al. (2021)]. Even though the name of the model is DeiT-tiny, it is not a small model. The model has 5M parameters, 12 transformer layers, 4 attention heads per layer, and the model dimension of 192. Both the WikiText-103 and ImageNet benchmarks require training the models on 4 NVIDIA 3090Ti GPUs for 3-4 days.
>
> Second, as the reviewer suggests, **we have conducted additional experiments** on different benchmarks to justify the advantage of our FourierFormer over the baseline softmax transformer. These additional benchmarks include **IWSLT’ 14 De-En machine translation task** [Cettolo et al. (2014)] and a set of 10 multivariate datasets from the **UEA Time Series Classification Archive** [Bagnall et al. (2018)]. We observe that our FourierFormers outperform the baseline softmax transformers significantly on both of these benchmarks. We summarize our results below and in Table 7 and 8 in Appendix E.4 and E.5, respectively, of the revision.
>
> Table 1: The FourierFormer vs. the baseline softmax transformer on the UEA Time Series Classification Archive benchmark
> | Dataset/Model       | *Baseline softmax (small)*          | Fourierformer   |
> | :---        |    :----:   |    :----:   |
> | EthanolConcentration     |   32.08 (33.70)  |   **36.12**    |
> | FaceDetection     |    68.70 (68.10)   |    **68.71**   |
> | HandWriting    |    **32.08 (30.50)**    |   31.68    |
> | HeartBeat    |    75.77 (77.60)    |    **76.42**    |
> | JapaneseVowels    |    **99.46 (99.40)**  |   99.37   |
> | PEMS-SF    |    82.66 (82.10)    |   **86.70**   |
> | SelfRegulationSCP1    |    91.46 (92.50)  |   **91.70**   |
> | SelfRegulationSCP2   |    54.72 (53.90)   |   **55.37**    |
> | SpokenArabicDigits   |    **99.33 (99.30)**    |   99.00  |
> | UWaveGestureLibrary   |    84.45 (85.60)    |  **86.66**  |
> | **Average Accuracy**   |   72.07 (72.27)    |  **73.17**   |
>
> Table 2: The FourierFormer vs. the baseline softmax transformer on the IWSLT’14 De-En machine translation benchmark
> | Method      | BLEU score
> | :---        |    :----:   |
> | *Baseline softmax*    |   34.42  |
> | FourierFormer     |    **34.68**   |
>
> Also, note that even though we refer to the baselines as the softmax transformers, those baseline models for different benchmarks are different. They use the same softmax attention. In our experiments, we replace this softmax attention in these baselines by the Fourier attention to form the corresponding FourierFormers.

---

> > ### Author Response · Authors · 2022-08-02
> > **Response to Reviewer Ky7B (2)**
> >
> > In addition to new results on more benchmarks, we have investigated the efficiency of the FourierFormer. We have included quantitative results on the runtime and GPU memory usage of the FourierFormer versus the baseline softmax transformer in Table 6 in Appendix E.3  of our revision. We summarize our results below. We observe that the runtime of the FourierFormer is slightly more than the runtime of the softmax transformer but can be further reduced with more optimized implementation. The GPU memory usage of the FourierFormer and the baseline softmax transformer are similar.
> >
> > Table 3: Runtime and GPU memory usage of the FourierFormer vs. the baseline softmax transformer. Both models are trained for the WikiText-103 language modeling task
> >
> > | Model       | Runtime (ms/sample)  (Train)    | GPU Memory (GB) (Train)    |Runtime (ms/sample)  (Test)    | GPU Memory (GB) (Test)    |
> > | :---        |    :----:   |    :----:   | :----:   |    :----:   |
> > | *Baseline softmax*       |    5.41   |     1.43    |     1.53   |     0.94    |
> > | FourierFormer    |    6.00    |      1.43    |     1.70   |     0.94    |
> >
> >
> > **References**
> >
> > [1] Imanol Schlag et al. Linear Transformers are Secretly Fast Weight Programmers. ICML, 2021.
> >
> > ​​[2] Jacob Devlin et al. BERT: Pre-training of Deep Bidirectional Transformers for Language Modeling. NAACL, 2019.
> >
> > [3] Edouard Grave et al.  Improving neural language models with a continuous cache. ICLR, 2017.
> >
> > [4] Yann N. Dauphin et al. Language modeling with gated convolutional networks. ICML, 2017.
> >
> > [5] Stephen Merity et al. An analysis of neural language modeling at multiple scales. arXiv, abs/1803.08240, 2018.
> >
> > [6] Jack W. Rae et al. Fast parametric learning with activation memorization. ICML, 2018.
> >
> > [7] Hugo Touvron et al. Training data-efficient image transformers & distillation through attention. ICML, 2021.
> >
> > [8] Mauro Cettolo et al. Report on the 11th iwslt evaluation campaign, iwslt 2014. In Proceedings of the International Workshop on Spoken Language Translation, Hanoi, Vietnam, volume 57, 2014.
> >
> > [9] Anthony Bagnall et al. The UEA multivariate time series classification archive, 2018. arXiv preprint arXiv:1811.00075.

---

> > > ### Author Response · Authors · 2022-08-02
> > > **Response to Reviewer Ky7B (3)**
> > >
> > > **Q2.In the paper, the authors say ‘However, equations (1) and (2) imply that the dot-product attention assumes the features (qi1, . . . , qiD) in qi, as well as the features (kj1, . . . , qjD) in kj , are independent’. This description also implies that v is not included in the consideration of the correlation. However, even it's straightforward to consider q, k as independent variables and v as dependent variables, in my opinion, q, k, and v don’t have a clear statistical relationship in Transformer, partially due to its poor interpretability. Hence, according to the author's idea, v may also be part of considering the correlation.**
> > >
> > > **Reply:** Under our nonparametric regression interpretation of self-attention, the attention keys $k_j$ and the attention queries $q_i$, $i,j=1,...,N$ are considered to be training and test data in a nonparametric regression problem, respectively. In Section 2 of our paper, we show that the self-attention implements the Nadaraya-Watson estimator by constructing a kernel function $\varphi_{\sigma}(q_i, k_j)$ between $q_i$ and $k_j$ where $\varphi_{\sigma}(.)$ is the isotropic multivariate Gaussian density function with diagonal covariance matrix. This implies that the features $(q_{i1}, . . . , q_{iD})$ in $q_i$ and the features $(k_{j1}, . . . , q_{jD})$ in $k_j$  are independent.
> > >
> > > Also, from our nonparametric regression interpretation of self-attention, the attention values $v_j$, $j=1,...,N$ are considered to be training targets. The features $(v_{j1}, . . . , v_{jD})$ in $v_j$ can be dependent or independent, and are not part of the correlated structure we consider in our paper. However, as the reviewer suggests, it is interesting to incorporate the dependency between features of $v_j$ into our nonparametric regression framework for self-attention, and we will explore it in the future work.
> > >
> > > -----
> > >
> > > We hope we have cleared your concerns about our work. We have also revised our manuscript according to your comments, and we would appreciate it if we can get your further feedback at your earliest convenience.

---

> > > > ### Author Response · Authors · 2022-08-06
> > > > **Response to Reviewer Ky7B - New Empirical Results to Further Address Q1**
> > > >
> > > > Regarding the concern from the reviewer that the experiments in our paper are no sufficient to demonstrate the effectiveness of the FourierFormer, in addition to experiments on IWSLT’ 14 De-En machine translation task [Cettolo et al. (2014)] and the UEA Time Series Classification Archive [Bagnall et al. (2018)] in our previous reply to Q1, as well as on the ImageNet image classification task and the WikiText-103 language modeling task in our paper, we have run additional experiments on the **continuous control tasks from the D4RL benchmark** [Fu et al. (2020)] to evaluate the performance of the FourierFormer on the offline reinforcement learning. On this benchmark, our decision FourierFormer significantly outperforms the baseline decision transformer [Chen et al.(2021)] on 8 out of 9 tasks and on average across tasks. Here, the decision FourierFormer is the decision transformer with the Fourier attention instead of the softmax attention. This result shows that our Fourier attention can be used in different transformer architectures to improve their performance. We include our results below and in Table 9 in Appendix E.7 of our revision. Each experiment result is averaged over 5 runs with different random seeds.
> > > >
> > > > We would appreciate it if you could let us know if our responses have addressed your concerns and whether you still have any other questions on the current draft and our rebuttal. We would be happy to do any follow-up discussion or address any additional comments.
> > > >
> > > >
> > > >
> > > > Table 4: The decision FourierFormer vs. the baseline decision transformer on the continuous control tasks from D4RL benchmark
> > > >
> > > > | Environment/Model       | Baseline decision transformer          | Decision FourierFormer   |
> > > > | :---        |    :----:   |    :----:   |
> > > > |         |    Medium-Expert   |     |
> > > > | HalfCheetah     |   91.03 (83.80)  |   **92.27**   |
> > > > | Hopper     |    110.30 (104.40)   |    **111.10**  |
> > > > | Walker   |    108.70 (107.70)    |    **108.9**   |
> > > > |         |    Medium-Replay   |     |
> > > > | HalfCheetah     |   35.31 (34.60)  |  **38.47**    |
> > > > | Hopper     |   85.61 (79.70)  |  **89.70**    |
> > > > | Walker     |   **66.11 (62.90)**  |  63.19    |
> > > > |         |    Medium   |     |
> > > > | HalfCheetah     |   42.28 (42.40)  |   **42.38**   |
> > > > | Hopper     |    61.47 (64.20)   |   **64.77**   |
> > > > | Walker   |    68.68 (70.60)    |     **70.42**    |
> > > > | **Avg Reward**   |   74.39 (72.20)    |  **75.69**  |
> > > >
> > > > **References**
> > > >
> > > > [1] Mauro Cettolo et al. Report on the 11th iwslt evaluation campaign, iwslt 2014. In Proceedings of the International Workshop on Spoken Language Translation, Hanoi, Vietnam, volume 57, 2014.
> > > >
> > > > [2] Anthony Bagnall et al. The UEA multivariate time series classification archive. arXiv preprint arXiv:1811.00075, 2018.
> > > >
> > > > [3] Justin Fu et al. D4rl: Datasets for deep data-driven reinforcement learning. arXiv preprint arXiv:2004.07219, 2020.
> > > >
> > > > [4] Lili Chen et al. Decision Transformer: Reinforcement Learning via Sequence Modeling. NeurIPS, 2021.

---

> > > > > ### Comment · Reviewer_Ky7B · 2022-08-08
> > > > > **Feedback**
> > > > >
> > > > > Thanks for the comprehensive response.
> > > > > In light of the author's clarifications and willingness to perform more experiments on benchmarks, I will raise my scores.
> > > > > However, the improvement is limited from current experiments (including the latest update). Meanwhile, I was wondering about the difficulty of hyperparameter-tuning on real-world datasets. Could you please discuss it on the rolling update of D4RL benchmark?

---

> > > > > > ### Author Response · Authors · 2022-08-08
> > > > > > **Thanks for your endorsement!**
> > > > > >
> > > > > > Thanks for your further feedback and we appreciate your endorsement. On the D4RL benchmark, like with other tasks studied in our paper, compared to the baseline, we only need to choose the function $\phi$ and the initial value for the learnable scalar $R$, i.e. $R_{\text{init}}$. In our D4RL experiments, we choose $\phi = x^{4}$ and $R_{\text{init}}=1$. Across our experiments, we observe that  $\phi = x^{2}$ or $x^{4}$ and $R_{\text{init}}=1$ or $2$ consistently yield good results. We have included the hyperparameter-tuning of the FourierFormer on the D4RL benchmark in Appendix E.7 of our revision.

---

### Official Review · Reviewer_XT54 · 2022-07-11

**Rating:** 8
**Confidence:** 4
**Soundness:** 4 excellent
**Presentation:** 4 excellent
**Contribution:** 4 excellent

**Summary:**

This paper proposes an alternative to softmax attention using the sinc function. The authors were well-motivated using the sinc function from the Fourier integral estimator and provided theoretical support for approximation error. They have done experiments on two datasets showing improvement over softmax attention.

**Questions:**

1. What is the connection and differences between the proposed sinc function-based attention and the Fnet [1], which uses Fourier transform instead of softmax attention?
2. Eq. 6. is derived upon considering only $\mathbf{v}$ and $\mathbf{k}$. How is it justified to replace $\mathbf{k}$ with $\mathbf{q}$ in line 119?
3. What was the rationale and practical reason to consider $\mathbf{k}_j$ s are i.i.d? In practice, when they went through an MLP, the channels became dependent on each other.

[1] Lee-Thorp, James, et al. "Fnet: Mixing tokens with Fourier transforms." arXiv preprint arXiv:2105.03824 (2021).


**Limitations:**

Authors have not adequately commented on the known limitations.

**Strengths And Weaknesses:**

Strengths:
1. The paper is a pleasant read and clear to understand.
2. The established connection between non-parametric kernel density estimator and self-attention.
3. The authors have provided well-motivated intuition for their proposed approach.
4. Related work has been moderately covered.
5. The evaluation is convincing to show the benefit of sinc-based attention.

Weaknesses:
1. The authors only experimented with one choice of $\phi$. It would be great to see what other suitable candidate of $\phi$ is possible.
2. How do they determine the value of $R$? Is it dataset-specific?
3. There are no quantitative results on the runtime of the proposed attention mechanism.

---

> ### Author Response · Authors · 2022-08-02
> **Response to Reviewer XT54 (1)**
>
> Thank you for your thoughtful review and valuable feedback. Below we address your concerns.
>
> -----
> **Q1. The authors only experimented with one choice of $\phi$. It would be great to see what other suitable candidate of $\phi$ is possible.**
>
> **Reply:** Thanks for your suggestion. We were inspired by the well-known Fejer-de la Vallee Poussin and the Jackson-de le Vallee Poussin kernels and chose $\phi(x) = x^{2m}$ where m is a positive integer. We have also tried other choices of $\phi$ that are nonnegative functions such as  $\phi(x) = |x|$, $\text{ReLU}(x)$, and $\text{sigmoid}(x)$. Those functions yield worse results than $\phi(x) = x^{2m}$. We have updated Table 4 in Appendix E.1 to include these new results.
>
> **Q2. How do they determine the value of R? Is it dataset-specific?**
>
> **Reply:** In FourierFormer, we make R a parameter that is learned from the data during the training of the model. We have discussed the effect of the initialization of R in Appendix E.2 and Table 5. We observe that when R is initialized too small (e.g. R_init = 0.1) or too big (e.g. R_init = 4), the accuracy of the trained FourierFormer decreases. R_init = 1, 2, 3 yields best results across tasks.
>
> **Q3. There are no quantitative results on the runtime of the proposed attention mechanism.**
>
> **Reply:** We have included quantitative results on the runtime and GPU memory usage of the FourierFormer versus the baseline softmax transformer in Table 6 in Appendix E.3  of our revision. We summarize our results below. We observe that the runtime of the FourierFormer is slightly more than the runtime of the softmax transformer but can be further reduced with a more optimized implementation. The GPU memory usage of the FourierFormer and the baseline softmax transformer are similar.
>
> Table 1: Runtime and GPU memory usage of the FourierFormer vs. the baseline softmax transformer. Both models are trained for the WikiText-103 language modeling task
>
> | Model       | Runtime (ms/sample)  (Train)    | GPU Memory (GB) (Train)    |Runtime (ms/sample)  (Test)    | GPU Memory (GB) (Test)    |
> | :---        |    :----:   |    :----:   | :----:   |    :----:   |
> | *Baseline softmax*       |    5.41   |     1.43    |     1.53   |     0.94    |
> | FourierFormer    |    6.00    |      1.43    |     1.70   |     0.94    |
>
> **Q4. What is the connection and differences between the proposed sinc function-based attention and the Fnet [1], which uses Fourier transform instead of softmax attention?
> [1] Lee-Thorp, James, et al. "Fnet: Mixing tokens with Fourier transforms." arXiv preprint arXiv:2105.03824 (2021).**
>
> **Reply:** The FNet in [Lee-Thorp et al. (2021)] replaces the self-attention layer in transformers with a Fourier layer, which applies a 2-D Discrete Fourier Transform to the input sequence in order to mix the elements of the input sequence along the sequence dimension and the hidden/feature dimension to form higher order units. In contrast, our proposed sinc function-based attention is based on the Fourier Integral Theorem, which is a combination of the Fourier transform and the inverse Fourier transform to estimate any real-value integrable function in R^{D}. Using the Fourier Integral Theorem, we develop our generalized Fourier density estimator. We then apply this generalized Fourier density estimator to the nonparametric regression problem in self-attention derived in Section 2 in our paper to develop the Fourier attention that can capture the correlation between features in the attention queries and keys.
>
> **References**
>
> [1] James Lee-Thorp et al. "Fnet: Mixing tokens with Fourier transforms." arXiv preprint arXiv:2105.03824 (2021).

---

> > ### Author Response · Authors · 2022-08-02
> > **Response to Reviewer XT54 (2)**
> >
> > **Q5. Eq. 6. is derived upon considering only v  and k. How is it justified to replace k with q in line 119?**
> >
> > **Reply:** In Eq. 6, the attention keys $k_j$ and the attention values $v_j$, $j=1,...,N$ can be considered as training data and training target. We use Eq. 6 to construct a kernel regression estimator of the function $f$ that map from new data $k$, i.e. test data, to the corresponding target, i.e. test target. As mentioned at the beginning of Section 2 in our paper, the attention queries $q_i$, $i=1,...,N$, can be considered as new input data. Thus, we replace $k$ in Eq. 6 with $q_i$.
> >
> > **Q6. What was the rationale and practical reason to consider k s are i.i.d? In practice, when they went through an MLP, the channels became dependent on each other.**
> >
> > **Reply:** We agree with the reviewer that in a multilayer transformer, $k_j$, $j=1,...,N$, might not be i.i.d. due to the potential use of an MLP that mixes tokens together before the self-attention layer. In our paper, we only focus on a single self-attention layer and  assume that $k_j$ are i.i.d., i.e. there is no MLP that mixes tokens together before the self-attention layer, to simplify the derivation of the nonparametric regression model underlying the self-attention and its analysis.
> >
> > -----
> >
> > We hope we have cleared your concerns about our work. We have also revised our manuscript according to your comments, and we would appreciate it if we can get your further feedback at your earliest convenience.

---

> > > ### Author Response · Authors · 2022-08-06
> > > **Response to Reviewer XT54 - Any further questions on our current draft**
> > >
> > > We would like to thank you again for your thoughtful reviews and valuable feedback.
> > >
> > > We would appreciate it if you could let us know if our responses have addressed your concerns and whether you still have any other questions on the current draft and our rebuttal.
> > >
> > > We would be happy to do any follow-up discussion or address any additional comments.

---

### Author Response · Authors · 2022-08-03
**General Response (1)**

Dear AC and reviewers,

Thanks for your thoughtful reviews and valuable comments, which have helped us improve the paper significantly. We are encouraged by the endorsements that: 1) The connection between non-parametric kernel density estimator and self-attention is established (Reviewer XT54); 2) Our ideas are novel (Reviewer Ky7B, MHeF, WPHc), and the mathematical arguments are complete (Reviewer Ky7B, MHeF) and ingenious (Reviewer Ky7B); 3) The experimental results verify the benefits of the proposed Fourier Former (Reviewer XT54, WPHc). We have updated our submission based on the reviewers' feedback, and we have highlighted our revision in blue.

One of the main concerns from the reviewers is that the experimental results in the paper are not sufficient to demonstrate the advantage of our proposed FourierFormer. We first address this comment here.

First, both WikiText-103 and ImageNet benchmarks that we use to verify the advantage of our FourierFormer are considered large-scale datasets. In particular, WikiText-103 consists of 28K training articles with 103M running words, and ImageNet consists of 1.28M training images. We also use strong baselines for both the WikiText-103 language modeling and the ImageNet image classification task. Particularly, for the WikiText-103 benchmark, our baseline model  with the medium configuration [Schlag et al. (2021)] reported in Table 1 in the paper has 90M parameters, 16 layers, 8 attention heads per layer, and hidden size of 256. The size of our baseline model is close to BERTBase [Devlin et al. (2019)], which has 110M parameters, 12 layers, 12 attention heads per layer, and hidden size of 768. Note that the baseline transformer we used is deeper than BERTBase. This baseline transformer attains a test perplexity (PPL) of 29.60 as reported in [Schlag et al. (2021)], which, on this WikiText-103 task, is better than or equivalent to popular transformer models including  [Grave et al. (2017)], [Dauphin et al. (2017)], [Merity et al. (2018)] and [Rae et al. (2018)]. For the ImageNet benchmark, we use the DeiT-tiny model in [Touvron et al. (2021)]. Even though the name of the model is DeiT-tiny, it is not a small model. The model has 5M parameters, 12 transformer layers, 4 attention heads per layer, and the model dimension of 192. Both the WikiText-103 and ImageNet benchmarks require training the models on 4 NVIDIA 3090Ti GPUs for 3-4 days.

---

> ### Author Response · Authors · 2022-08-03
> **General Response (2)**
>
> Second, as the reviewer suggests, **we have conducted additional experiments** on different benchmarks to justify the advantage of our FourierFormer over the baseline softmax transformer. These additional benchmarks include **IWSLT’ 14 De-En machine translation task** [Cettolo et al. (2014)], a set of 10 multivariate datasets from the **UEA Time Series Classification Archive** [Bagnall et al. (2018)], and the **continuous control tasks from the D4RL benchmark** [Fu et al. (2020)]. We observe that our FourierFormers outperform the baseline softmax transformers significantly on these benchmarks. We summarize our results below and in Table 7, 8, and 9 in Appendix E.4, E.5, and E7, respectively, of the revision.
>
> Table 1: The FourierFormer vs. the baseline softmax transformer on the UEA Time Series Classification Archive benchmark
> | Dataset/Model       | *Baseline softmax (small)*          | Fourierformer   |
> | :---        |    :----:   |    :----:   |
> | EthanolConcentration     |   32.08 (33.70)  |   **36.12**    |
> | FaceDetection     |    68.70 (68.10)   |    **68.71**   |
> | HandWriting    |    **32.08 (30.50)**    |   31.68    |
> | HeartBeat    |    75.77 (77.60)    |    **76.42**    |
> | JapaneseVowels    |    **99.46 (99.40)**  |   99.37   |
> | PEMS-SF    |    82.66 (82.10)    |   **86.70**   |
> | SelfRegulationSCP1    |    91.46 (92.50)  |   **91.70**   |
> | SelfRegulationSCP2   |    54.72 (53.90)   |   **55.37**    |
> | SpokenArabicDigits   |    **99.33 (99.30)**    |   99.00  |
> | UWaveGestureLibrary   |    84.45 (85.60)    |  **86.66**  |
> | **Average Accuracy**   |   72.07 (72.27)    |  **73.17**   |
>
> Table 2: The FourierFormer vs. the baseline softmax transformer on the IWSLT’14 De-En machine translation benchmark
> | Method      | BLEU score
> | :---        |    :----:   |
> | *Baseline softmax*    |   34.42  |
> | FourierFormer     |    **34.68**   |
>
> Table 3: The decision FourierFormer vs. the baseline decision transformer on the continuous control tasks from D4RL benchmark
>
> | Environment/Model       | Baseline decision transformer          | Decision FourierFormer   |
> | :---        |    :----:   |    :----:   |
> |         |    Medium-Expert   |     |
> | HalfCheetah     |   91.03 (83.80)  |   **92.27**   |
> | Hopper     |    110.30 (104.40)   |    **111.10**  |
> | Walker   |    108.70 (107.70)    |    **108.9**   |
> |         |    Medium-Replay   |     |
> | HalfCheetah     |   35.31 (34.60)  |  **38.47**    |
> | Hopper     |   85.61 (79.70)  |  **89.70**    |
> | Walker     |   **66.11 (62.90)**  |  63.19    |
> |         |    Medium   |     |
> | HalfCheetah     |   42.28 (42.40)  |   **42.38**   |
> | Hopper     |    61.47 (64.20)   |   **64.77**   |
> | Walker   |    68.68 (70.60)    |     **70.42**    |
> | **Avg Reward**   |   74.39 (72.20)    |  **75.69**  |
>
> Also, note that even though we refer to the baselines as the softmax transformers, those baseline models for different benchmarks are different. They use the same softmax attention. In our experiments, we replace this softmax attention in these baselines by the Fourier attention to form the corresponding FourierFormers.

---

> > ### Author Response · Authors · 2022-08-06
> > **General Response (3)**
> >
> > In addition to new results on more benchmarks, we have investigated the efficiency of the FourierFormer. We have included quantitative results on the runtime and GPU memory usage of the FourierFormer versus the baseline softmax transformer in Table 6 in Appendix E.3  of our revision. We summarize our results below. We observe that the runtime of the FourierFormer is slightly more than the runtime of the softmax transformer but can be further reduced with more optimized implementation. The GPU memory usage of the FourierFormer and the baseline softmax transformer are similar.
> >
> > Table 4: Runtime and GPU memory usage of the FourierFormer vs. the baseline softmax transformer. Both models are trained for the WikiText-103 language modeling task
> >
> > | Model       | Runtime (ms/sample)  (Train)    | GPU Memory (GB) (Train)    |Runtime (ms/sample)  (Test)    | GPU Memory (GB) (Test)    |
> > | :---        |    :----:   |    :----:   | :----:   |    :----:   |
> > | *Baseline softmax*       |    5.41   |     1.43    |     1.53   |     0.94    |
> > | FourierFormer    |    6.00    |      1.43    |     1.70   |     0.94    |
> >
> > **References**
> >
> > [1] Imanol Schlag et al. Linear Transformers are Secretly Fast Weight Programmers. ICML, 2021.
> >
> > ​​[2] Jacob Devlin et al. BERT: Pre-training of Deep Bidirectional Transformers for Language Modeling. NAACL, 2019.
> >
> > [3] Edouard Grave et al.  Improving neural language models with a continuous cache. ICLR, 2017.
> >
> > [4] Yann N. Dauphin et al. Language modeling with gated convolutional networks. ICML, 2017.
> >
> > [5] Stephen Merity et al. An analysis of neural language modeling at multiple scales. arXiv, abs/1803.08240, 2018.
> >
> > [6] Jack W. Rae et al. Fast parametric learning with activation memorization. ICML, 2018.
> >
> > [7] Hugo Touvron et al. Training data-efficient image transformers & distillation through attention. ICML, 2021.
> >
> > [8] Mauro Cettolo et al. Report on the 11th iwslt evaluation campaign, iwslt 2014. In Proceedings of the International Workshop on Spoken Language Translation, Hanoi, Vietnam, volume 57, 2014.
> >
> > [9] Anthony Bagnall et al. The UEA multivariate time series classification archive, 2018. arXiv preprint arXiv:1811.00075.
> >
> > [10] Justin Fu et al. D4rl: Datasets for deep data-driven reinforcement learning. arXiv preprint arXiv:2004.07219, 2020.
> >
> >
> > -----
> >
> > We are glad to answer any further questions you have on our submission.

---

### Author Response · Authors · 2022-08-03
**Summary of Revision**

Incorporating the comments and suggestions from all reviewers, besides fixing typos and notations, we have made the following main changes in the revised paper.

1. **We have added additional experiments** on the **IWSLT’ 14 De-En machine translation task** [Cettolo et al. (2014)], a set of 10 multivariate datasets from the **UEA Time Series Classification Archive** [Bagnall et al. (2018)] and the **continuous control tasks from D4RL benchmark** [Fu et al. (2020)] for offline reinforcement learning to justify the advantage of our FourierFormer. We summarize our results in Table 7, 8 and 9 in Appendix E.4, E.5 and E.7, respectively, of the revision.

2. We have included quantitative results on the runtime and GPU memory usage of the FourierFormer versus the baseline softmax transformer in Table 6 in Appendix E.3 of our revision

3. We have added a background section on the nonparametric kernel regression, kernel density estimation, and the Fourier Integral theorem in Appendix A of the revision.

4. We have conducted additional synthetic experiments to empirically justify Theorems 1 and 2. We have described our experiments and summarized our results in Section E.6, Figure 1 and 2 in Appendix E of the revision.

5. We have updated Table 4 in Appendix E.1 to study the effect using other function $\phi(\cdot)$ rather than $\phi(x) = x^{2m}$ where m is a positive integer. In particular, we consider $\phi(x) = |x|$, $\text{ReLU}(x)$, and $\text{sigmoid}(x)$.

6. We have provided an explanation for the symbol annotations $s$ and $R$ right after Eq. 9 and further explained the derivation of the Fourier Integral Theorem in Appendix A.3 of our revision.

**References**

[1] Mauro Cettolo et al. Report on the 11th iwslt evaluation campaign, iwslt 2014. In Proceedings of the International Workshop on Spoken Language Translation, Hanoi, Vietnam, volume 57, 2014.

[2] Anthony Bagnall et al. The uea multivariate time series classification archive. arXiv preprint arXiv:1811.00075, 2018.

[3] Justin Fu et al. D4rl: Datasets for deep data-driven reinforcement learning. arXiv preprint arXiv:2004.07219, 2020.

---

### Author Response · Authors · 2022-08-06
**Any Questions from the Reviewers? New Empirical Results on the D4RL Continuous Control Tasks**

Dear reviewers,

We would like to thank all reviewers again for your thoughtful reviews and valuable feedback. We have just updated our manuscript and added new replies to your comments and questions with our latest experimental results. We have summarized the changes we made in the manuscript in the Summary of Revision below.

We would appreciate it if you could let us know if there are additional questions or concerns about our revision and rebuttal. We would be happy to do any follow-up discussion or address any additional comments.

To further verify the advantage of the FourierFormer, in addition to the experiments on the IWSLT’ 14 De-En machine translation task [Cettolo et al. (2014)] and the UEA Time Series Classification Archive [Bagnall et al. (2018)] in our previous reply to Q1, as well as on the ImageNet image classification task and the WikiText-103 language modeling task in our paper, we have run additional experiments on the **continuous control tasks from the D4RL benchmark** [Fu et al. (2020)] to evaluate the performance of the FourierFormer on the offline reinforcement learning. On this benchmark, our decision FourierFormer significantly outperforms the baseline decision transformer [Chen et al.(2021)] on 8 out of 9 tasks and on average across tasks. Here, the decision FourierFormer is the decision transformer with the Fourier attention instead of the softmax attention. This result shows that our Fourier attention can be used in different transformer architectures to improve their performance. We include our results below and in Table 9 in Appendix E.7 of our revision. Each experiment result is averaged over 5 runs with different random seeds.

Table for decision FourierFormer vs. the baseline decision transformer on the continuous control tasks from D4RL benchmark

| Environment/Model       | Baseline decision transformer          | Decision FourierFormer   |
| :---        |    :----:   |    :----:   |
|         |    Medium-Expert   |     |
| HalfCheetah     |   91.03 (83.80)  |   **92.27**   |
| Hopper     |    110.30 (104.40)   |    **111.10**  |
| Walker   |    108.70 (107.70)    |    **108.9**   |
|         |    Medium-Replay   |     |
| HalfCheetah     |   35.31 (34.60)  |  **38.47**    |
| Hopper     |   85.61 (79.70)  |  **89.70**    |
| Walker     |   **66.11 (62.90)**  |  63.19    |
|         |    Medium   |     |
| HalfCheetah     |   42.28 (42.40)  |   **42.38**   |
| Hopper     |    61.47 (64.20)   |   **64.77**   |
| Walker   |    68.68 (70.60)    |     **70.42**    |
| **Avg Reward**   |   74.39 (72.20)    |  **75.69**  |

**References**

[1] Mauro Cettolo et al. Report on the 11th iwslt evaluation campaign, iwslt 2014. In Proceedings of the International Workshop on Spoken Language Translation, Hanoi, Vietnam, volume 57, 2014.

[2] Anthony Bagnall et al. The UEA multivariate time series classification archive. arXiv preprint arXiv:1811.00075, 2018.

[3] Justin Fu et al. D4rl: Datasets for deep data-driven reinforcement learning. arXiv preprint arXiv:2004.07219, 2020.

[4] Lili Chen et al. Decision Transformer: Reinforcement Learning via Sequence Modeling. NeurIPS, 2021.

---

### Meta-Review · Area_Chair_CeK4 · 2022-08-26

**Recommendation:** Accept
**Confidence:** Certain

**Metareview:**

Overall, the reviews about this paper are very positive. The authors spent great effort engaging in discussions and improving the paper with clarifications and additional experiments. We recommend accepting the paper.

**Award:**

No

---

### Decision · Program_Chairs · 2022-09-14

Accept